# Left Atrial Diameter in the Prediction of Thromboembolic Event and Death in Atrial Fibrillation

**DOI:** 10.3390/jcm11071838

**Published:** 2022-03-26

**Authors:** Rungroj Krittayaphong, Arjbordin Winijkul, Poom Sairat

**Affiliations:** Division of Cardiology, Department of Medicine, Faculty of Medicine Siriraj Hospital, Mahidol University, Bangkok 10700, Thailand; arjbordin_winijkul@yahoo.com (A.W.); poom.kaab@gmail.com (P.S.)

**Keywords:** incremental prognostic value, left atrial size, thromboembolism, patients, non-valvular atrial fibrillation

## Abstract

Background: This study aimed to determine the predictive value of left atrial diameter (LAD), and the incremental prognostic value of LAD in combination with CHA_2_DS_2_-VASc score for predicting thromboembolic event and all-cause death in patients with non-valvular atrial fibrillation (AF). Methods: This is a prospective study from 27 hospitals during 2014–2017. LADi is LAD data indexed by body surface area, and LADi in the 4th quartile (LADi Q4) was considered high. Results: A total of 2251 patients (mean age 67.4 years, 58.6% male) were enrolled. Mean follow-up duration was 32.3 months. Rates of thromboembolic events and all-cause death were significantly higher in LADi Q4 patients than in LADi Q1–3 patients (2.89 vs. 1.11 per 100 person-years, *p* < 0.001, and 7.52 vs. 3.13 per 100 person-years, *p* < 0.001, respectively). LADi Q4 is an independent predictor of thromboembolic events and all-cause death with an adjusted hazard ratio and 95% confidence interval of 1.94 (1.24–3.05) and 1.81 (1.38–2.37), respectively. LADi has incremental prognostic value on top of the CHA_2_DS_2_-VASc score with the increase in global chi-square for thromboembolism (*p* = 0.005) and all-cause death (*p* < 0.001). Conclusions: LADi is an independent predictor of thromboembolic event and has incremental prognostic value in combination with CHA_2_DS_2_-VASc score in AF patients.

## 1. Introduction

Non-valvular atrial fibrillation (AF) is associated with an approximate five-times increased risk of ischemic stroke or systemic embolic complication compared to those without AF [1]. The CHA_2_DS_2_-VASc scoring system is a widely accepted and recommended tool for assessing thromboembolic risk [2]. The system has been validated in both Western [3] and Asian population [4]. The guidelines of the European Society of Cardiology (ESC) [5] and the American College of Cardiology (ACC) [6] recommend that patients at very low risk for thromboembolic complication as indicated by a CHA_2_DS_2_-VASc score of zero do not require oral anticoagulant. The benefit of anticoagulant outweighs the risk when the annual rate of thromboembolism exceeds 1.7% for vitamin K antagonist, and 0.9% for non-vitamin K antagonist oral anticoagulant (NOAC) [7]. Warfarin remains the preferred anticoagulant agent in many Asian countries [8,9,10]. The thromboembolic complication rate is higher in the Asian population than in the Western population [4,11]. Data from Taiwan showed that the rate of thromboembolism in patients with a CHA_2_DS_2_-VASc score of zero was greater than 1% [4]. Therefore, some patients with a CHA_2_DS_2_-VASc score of zero may benefit from OAC. A modified CHA_2_DS_2_-VASc scoring system has been proposed that decreases the age cutoff from 65 years to 50 years in males. It was reported that this modified system may be more suitable than the conventional CHA_2_DS_2_-VASc scoring system for assessing stroke risk, and for identifying patients with a CHA_2_DS_2_-VASc score of zero 0 who would benefit from OAC [12]. Other factors that have been proposed as potential predictors of thromboembolic risk include renal function, left atrial size, and certain biomarkers [13,14]. Left atrial diameter (LAD) was reported to be a significant predictor of thromboembolism independent of AF [15]. Although LAD has been shown to be an independent predictor of stroke [16] and recurrent stroke [17] in patients with AF, little is known about the incremental prognostic value of combination LAD and CHA_2_DS_2_-VASc scoring. The aim of this study was to determine the predictive value of left atrial diameter (LAD), and the incremental prognostic value of LAD in combination with CHA_2_DS_2_-VASc score for predicting thromboembolic event and all-cause death in patients with AF.

## 2. Methods

### 2.1. Study Population

The data including in this study is from a prospective registry entitled the cohort of antithrombotic use and Optimal INR Level in patients with non-valvular atrial fibrillation in the Thailand (COOL-AF) registry, which was established to collect data of patients with non-valvular AF from 27 large hospitals in Thailand. The enrollment period was during 2014–2017. Patients with non-valvular AF who were aged 18 years or older and who had available LAD data from echocardiogram were enrolled. The exclusion criteria were (1) prosthetic heart valve; (2) rheumatic valve disease; (3) ischemic stroke within 3 months; (4) AF from transient reversible cause; (5) hematologic disease that increased the risk of bleeding, such as thrombocytopenia or myeloproliferative disorders; (6) pregnancy; (7) inability to attend follow-up; (8) refusal to participate; and/or (9) life expectancy less than 3 years. The study was approved by Siriraj lnstitutional Review Board (COA no. Si 317/2014) and Central Research Ethics Committee (COA-CREC 003/2014). Each patient gave written informed consent before participation. This study was conducted in accordance with the principles set forth in the Declaration of Helsinki (1964) and all of its subsequent amendments, and all patients provided written informed consent to participate.

### 2.2. Study Protocol

All investigators were informed to enroll consecutive patients according to the inclusion and exclusion criteria. After the informed consent process, investigators collected all required data from the medical record and from patient interview. All data were written in a case record form. The data from the case record form was then input into a web-based data collection system. All case record forms were sent to a central data management site where all data were reentered and cross checked to ensure accuracy. Any questions that arose during the data verification process were sent to the appropriate investigator for confirmation or correction. Site monitoring was performed at every participating hospital to ensure compliance with all aspects of the study protocol. Patients were followed-up at 6, 12, 18, 24, 30, and 36 months. Data relating to cardiovascular events, clinical, laboratory, and medications were collected at each follow-up visit.

### 2.3. Data Collection

The following data were collected during the baseline visit: (1) demographic data; (2) weight and height; (3) vital signs; (4) duration and type of AF; (5) symptoms of AF; (6) comorbid conditions, such as hypertension, diabetes mellitus, dyslipidemia, and current smoker status; (7) history of coronary artery disease (CAD), heart failure (HF), ischemic stroke, and chronic kidney disease (CKD); (8) laboratory data; (9) other investigations, such as electrocardiogram (ECG) and echocardiogram; (10) antithrombotic medications, such as anticoagulants and antiplatelets; (11) rate or rhythm control medications; and (12) other medications. Each component of the CHA_2_DS_2_-VASc score (C = congestive heart failure; H = hypertension; A = age ≥ 75 years; D = diabetes mellitus; S = stroke or TIA; V = vascular disease; A = age 65–74 years; and Sc = female sex) and the HAS-BLED score (hypertension, abnormal liver/renal function, stroke history, bleeding history or predisposition, labile INR, elderly, drug/alcohol usage) was also recorded. LAD data were recorded from echocardiogram.

### 2.4. Left Atrial Diameter (LAD) Measurement

LAD was acquired from the measurement of the anteroposterior diameter of the left atrium in the end systole from the leading edge of the posterior aortic wall to the leading edge of the posterior left atrial wall in the parasternal long-axis view using transthoracic echocardiography [18]. LAD index (LADi) was calculated by adjusting LAD to the body surface area using the following formula: BSA (m^2^) = square root{[height (cm) × weight (kg)]/3600} [19].

### 2.5. Outcome Measurement

The follow-up visits were every 6 months until 3 years. The clinical outcomes were thromboembolic event (ischemic stroke, TIA, or systemic embolism), all-cause death, and thromboembolic event or all-cause death. Ischemic stroke was defined as a sudden onset of neurological deficit lasting at least 24 h caused by a disruption of blood flow to the brain. TIA was defined as a neurological deficit that lasted less than 24 h. All supporting documents for each clinical outcome were required to be uploaded into the web-based system. These documents were reviewed by the adjudication committee to verify the occurrence a clinical outcome in each reported case.

In order to minimize the bias, all outcomes were confirmed by a separate adjudication team. The sample size of this registry was enough to determine the differences in outcome between 2 groups with 90% power.

### 2.6. Statistical Analysis

The descriptive statistics used were mean plus/minus standard deviation for continuous data, and number and percentage for categorical data. Student’s *t*-test was used to compare normally distributed continuous data, and chi-square test was used to compare categorical data. The incidence rate of clinical outcomes is shown as the rate per 100 person–years. A comparison of the incidence rate between 2 groups was performed using the Poisson method. Cox proportional hazards model was used for univariate and multivariate analysis to determine the ability of LADi to independently predict the clinical outcomes given the time data and adjustment for age, gender, symptomatic AF, history of HF, history of CAD, cardiac implantable electronic device, history of ischemic stroke/TIA, hypertension, diabetes mellitus, smoking, dyslipidemia, renal replacement therapy, dementia, CKD, and history of bleeding, OAC, and antiplatelet. The predictive value of LADi was assessed using quartiles of LADi (Q1 = 1st quartile, Q2 = 2nd quartile, Q3 = 3rd quartile, and Q4 = 4th quartile), as follows: LADi Q1 = LADi < 23.48 mm/BSA, LADi Q2 = LADi 23.48–26.809 mm/BSA, LADi Q3 = LADi 26.81–30.51159 mm/BSA, and LADi Q4 = LA dimension ≥30.5116 mm/BSA. High LADi was defined as the top quartile of LADi or LADi Q4. Sensitivity analysis was performed for the assessment of prognostic value of LADi according to the classification of the American Society of Echocardiography [20]. The incremental prognostic value of LADi combined with the CHA_2_DS_2_-VASc score was assessed, as follows: (1) survival analysis of four groups of patients (LADi Q4 and CHA_2_DS_2_-VASc ≥ 2, LADi Q1–Q3 and CHA_2_DS_2_-VASc ≥ 2, LADi Q4 and CHA_2_DS_2_-VASc 0–1, and LADi Q1–Q3 and CHA_2_DS_2_-VASc 0–1); and (2) a comparison of the global chi-square derived from each hierarchical model from the Cox proportional hazards model. A *p*-value less than 0.05 was considered to reflect statistical significance. All statistical analyses were performed using SPSS Statistics software (SPSS, Inc., Chicago, IL, USA).

## 3. Results

### 3.1. Study Population

A total of 2251 patients (mean age: 67.4 ± 11.3 years, 58.6% male) were enrolled. A flow diagram of patient enrollment and group allocation is shown in Figure 1. The mean CHA_2_DS_2_-VASc and HAS-BLED scores were 3.1 ± 1.7 and 1.5 ± 1.0, respectively. The average LADi was 27.0 ± 5.9 mm/m^2^. Baseline characteristics of all patients and compared among the four LADi groups are shown in Table 1. Patients with high LADi (LADi Q4) were significantly older; had a greater proportion of females, permanent AF, a history of HF, CKD, OAC use, antiplatelet use, and cardiovascular medications; had significantly higher CHA_2_DS_2_-VASc and HAS-BLED scores; and, had a significantly longer duration of AF.

### 3.2. LADi, CHA_2_DS_2_-VASc Score, and Incidence Rate of Clinical Outcomes

The mean follow-up duration was 32.3 ± 8.2 months. During follow-up, there was a total of 90 thromboembolic events (4.0%), 247 all-cause deaths (11.0%), and 308 thromboembolic events or deaths (13.7%). Among the patients who experienced thromboembolic events, 87 had ischemic stroke or TIA, and 4 patients had systemic embolism (1 patient had both). The cause of death was cardiovascular in 76 (30.8%), non-cardiovascular in 121 (49.0%), and undetermined in 50 (20.2%) cases.

The incidence rate of thromboembolism was 1.53 (2.07–3.94) per 100 person-years. Figure 2 demonstrates the incidence rate according to combination CHA_2_DS_2_-VASc score and LADi for predicting thromboembolic event, all-cause death, and thromboembolic event or all-cause death. Figure 2 also shows the incidence rate according to CHA_2_DS_2_-VASc score 0–1 (low-risk group) or CHA_2_DS_2_-VASc score ≥ 2 (high-risk group) combined with LADi (right images) for predicting thromboembolic event, all-cause death, and thromboembolic event or all-cause death. That analysis showed LADi combined with CHA_2_DS_2_-VASc score to be helpful for predicting clinical outcomes in both the high-risk and low-risk groups.

### 3.3. LADi as a Predictor for Clinical Outcome: Multivariate Analysis

Figure 3 demonstrates an adjusted analysis to determine the predictive value of LADi Q4 for thromboembolic event, all-cause death, and combined outcomes according to patient profile, AF characteristics, comorbid conditions, and antithrombotic medications. That analysis showed high LADi Q4 to be an independent predictor of thromboembolic event (Figure 3A), all-cause death (Figure 3B), and thromboembolic event or all-cause death (Figure 3C) based on the adjusted hazard ratio and 95% CI.

### 3.4. LADi Confers Incremental Prognostic Value When Combined with CHA_2_DS_2_-VASc Score Predict Clinical Outcomes

Analysis of the incremental prognostic ratio using global chi-square test from Cox proportional hazards model demonstrated that LADi Q4 significantly improves the ability to predict the clinical outcomes when combined with the CHA_2_DS_2_-VASc score (Figure 4).

Figure 5 shows a forest plot of the adjusted hazard ratio for the clinical outcomes using patients with a CHA_2_DS_2_-VASc score 0–1 and LADi Q1–3 as the reference group. The magnitude of the increased risk as indicated by the hazard ratio shows the risk to be highest in patients with a CHA_2_DS_2_-VASc score ≥ 2 and LADi Q4, followed by CHA_2_DS_2_-VASc score 0–1 and LADi Q4, and CHA_2_DS_2_-VASc score ≥ 2 and LADi Q1–3. Figure 5 also shows hazard graphs of the clinical outcomes stratified by LADi group and CHA_2_DS_2_-VASc score. Patients with CHA_2_DS_2_-VASc score ≥ 2 and LADi Q4 had the highest risk for all clinical outcomes, and those with a CHA_2_DS_2_-VASc score 0–1 and LADi Q1–3 had the lowest risk. Patients who had only one of the two factors had intermediate risk for the clinical outcomes.

### 3.5. Sensitivity Analysis

Sensitivity analysis was performed and displayed as cubic spline graph for the assessment of predictive value of LADi on thromboembolic event, all-cause death by treating LADi as a continuous variable displayed as unadjusted and adjusted hazard ratio (Appendix A). The risk of thromboembolic event and all-cause death increased as LADi increased. We performed an interaction test to determine whether the predictive ability of LADi was different between patients taking and not taking OAC. The results showed no significant effect of OAC use on the predictive ability of LADi for all clinical outcomes.

We also performed sensitivity analysis to investigate the relationship between LADi and the risk of CV-related and non-CV-related death. Among the 247 deaths, 76 were CV-related deaths (30.8%), 121 were non-CV-related deaths (49.0%), and the cause of death in the remaining 20.2% of cases was undetermined. The incidence rates of CV-related and non-CV-related deaths significantly increased in patients with LADi Q4 compared to those with LADi Q1–3 (2.34 vs. 0.94 per 100 person-years, respectively; *p* < 0.001 for CV-related death, and 3.06 vs. 1.71 per 100 person-years, respectively; *p* = 0.002 for non-CV-related death). The difference between the LADi Q4 and the LADi Q1–Q3 groups was more pronounced among those who had experienced a CV-related death. Sensitivity analysis was performed using LADi severity classification of the American Society of Echocardiography. The results showed that the risk of thromboembolic event and all-cause death significantly increased as the LADI severity increase (Appendix A) which is in the similar direction with the classification by quartiles.

Additional analysis was performed to determine the influence of the use or non-use of OAC on the results of LADi on clinical outcomes. That analysis showed no significant effect of OAC use on the ability of LADi to independently predict a thromboembolic event, all-cause death, or thromboembolic event or death (interaction test *p*-value = 0.102, 0.476, and 0.141, respectively).

## 4. Discussion

The results of this prospective multicenter AF registry in Thailand indicate that LADi is a strong independent predictor for thromboembolic events and all-cause death. The ability of LADi to predict all 3 clinical outcomes was shown to be independent of baseline clinical data, including CHA_2_DS_2_-VASc score and antithrombotic drug use.

The CHA_2_DS_2_-VASc scoring system is a widely recommended, accepted, and used tool for assessing the risk of thromboembolic events in patients with non-valvular AF [2]. However, the reported C-statistics ranged from 0.65 to 0.7 [2,12,21], which suggests that factors not included in the CHA_2_DS_2_-VASc scoring system play a role in thromboembolism prediction. Other factors that were reported to be predictors of thromboembolism include biomarkers [22], renal function [13], and LADi [15,16,17]. Biomarkers that were shown to be good predictors include N-terminal brain natriuretic peptide (NT-proBNP) and growth-differentiation factor-15 (GDF-15), but these biomarkers are not routinely measured [22,23]. The addition of biomarker information increased the C-statistic up to 0.68 [22]. LADi is a simple measure that is derived from echocardiography, which is a commonly performed investigation in patients with AF.

The results of our study demonstrate that LADi Q4 is significantly associated with an increased risk of thromboembolism and all-cause death. When we categorized patients into two groups (CHA_2_DS_2_-VASc score 0–1 and CHA_2_DS_2_-VASc score ≥ 2), LADi increased the risk of thromboembolism twice in the low-to-intermediate risk group, and almost tripled the risk in the high-risk group. Some populations have an increased risk of AF-related thromboembolism. By way of example, among Taiwanese patients with a CHA_2_DS_2_-VASc score of zero, the annual stroke rate was 1.15% [4]. In this setting, the addition of LADi data will help to identify patients in the low-risk group that would benefit from OAC to prevent stroke since their stroke risk is above the threshold of benefit/risk balance, especially the threshold for NOACs, which is much lower than the threshold for warfarin [7].

The results of our multivariate analysis showed the predictive value of LADi to be strong and independent of other factors. The benefit of combining LADi with the CHA_2_DS_2_-VASc score to predict the clinical outcomes was shown to have incremental prognostic value as demonstrated by the change in global chi-square. The survival analysis by Cox proportional hazards model demonstrated that patients with a CHA_2_DS_2_-VASc score ≥ 2 and LADi Q4 had the highest risk, and that patients with a CHA_2_DS_2_-VASc score 0–1 and LADi Q1–3 had the lowest risk. This finding emphasizes that LADi data may help identify patients in the low-to-intermediate risk group who may benefit from OAC, and may help to improve time in therapeutic range (TTR) among patients taking warfarin, which is the most widely used OAC in Thai AF patients.

In addition to predicting the risk of thromboembolism, LADi was also shown to predict the risk of all-cause death. Although LADi may indicate a more advanced stage of heart disease or a more advanced AF status, the results from our study indicate that LADi remains a significant and independent predictor for all-cause death irrespective of AF type and comorbid conditions. The CHA_2_DS_2_-VASc scoring system, which was originally developed for use as a stroke risk prediction tool, has also been shown to be an effective and independent predictor for all-cause death [24]. The results of our study showed LADi Q4 to be an independent predictor of all-cause death independent of CHA_2_DS_2_-VASc score. A previous study demonstrated that patients who had left atrial enlargement had a higher CHA_2_DS_2_-VASc score compared to those with normal left atrial size and thromboembolic risk as assessed by the CHA_2_DS_2_-VASc score compared is associated with an increase in left atrial diameter [25]. A study in the Chinese population has shown that left atrial enlargement is one of the independent predictors for left atrial thrombus [26]. However, CHA_2_DS_2_-VASc score is not an independent predictor for left atrial thrombus.

The concept of atrial cardiopathy has been proposed as a potential cause of AF, and it can lead to thromboembolism due to left atrial function abnormality, even without AF [27]. Data from a large retrospective cohort from Canada indicated that, in the absence of AF, increased LAD significantly and independently increased the risk of ischemic stroke at 2 years with C-statistics ranging from 0.68 to 0.75 [28]. The combination of LAD and CHA_2_DS_2_-VASc score, even in patients without AF, significantly improved stroke risk prediction. Therefore, the concept that AF causes left atrial dilatation and stroke should be changed to a bidirectional association between increased LAD and AF, and each of the two factors can lead to stroke [29]. Whether OAC can prevent stroke in patients with atrial cardiopathy without AF remains still needs to be determined [30].

### Limitations

This study has some mentionable limitations. First, since this prospective registry collected data from 27 medium/large hospitals in Thailand, our results may not be generalizable to all AF population in Thailand or to AF patients in other care settings. Second, although we informed all investigators to enroll AF patients consecutively, some selection bias during the patient enrollment process cannot be ruled out. Third, there was a large number of patients that had to be excluded due to missing or incomplete LADi data. Reasons included missing echocardiographic data, the echocardiography report experienced age-related deterioration in the medical record, or echocardiography was performed at a different hospital before being referred to our center.

## 5. Conclusions

LADi Q4 is a strong and independent predictor of thromboembolism and all-cause death in patients with non-valvular AF, and it has incremental prognostic value when combined with CHA_2_DS_2_-VASc score. LADi data may help to guide the decision whether to prescribe or not prescribe OAC in the low-to-intermediate risk group and may help to improve TTR control among patients taking warfarin.

## Figures and Tables

**Figure 1 jcm-11-01838-f001:**
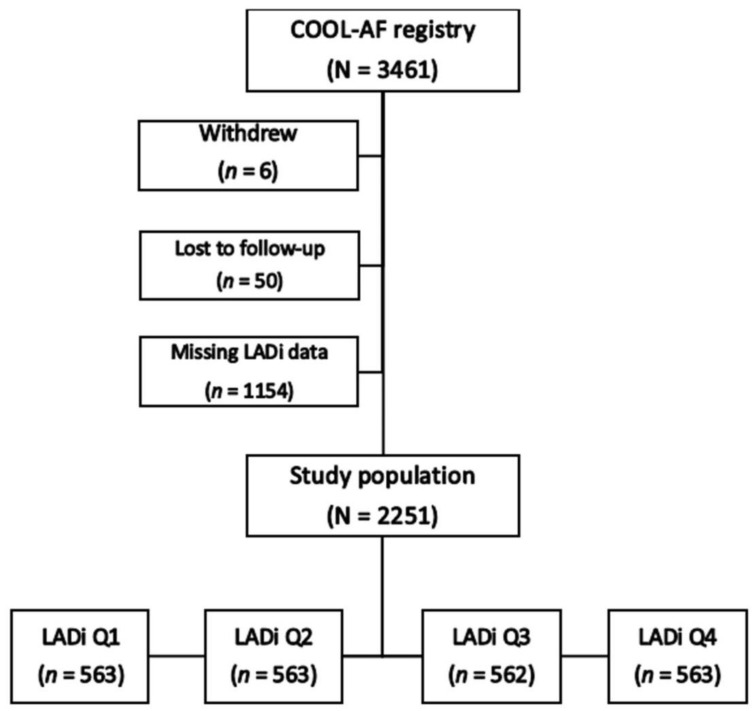
Flow diagram of study population (LADi = left atrial diameter index, Q = quartile).

**Figure 2 jcm-11-01838-f002:**
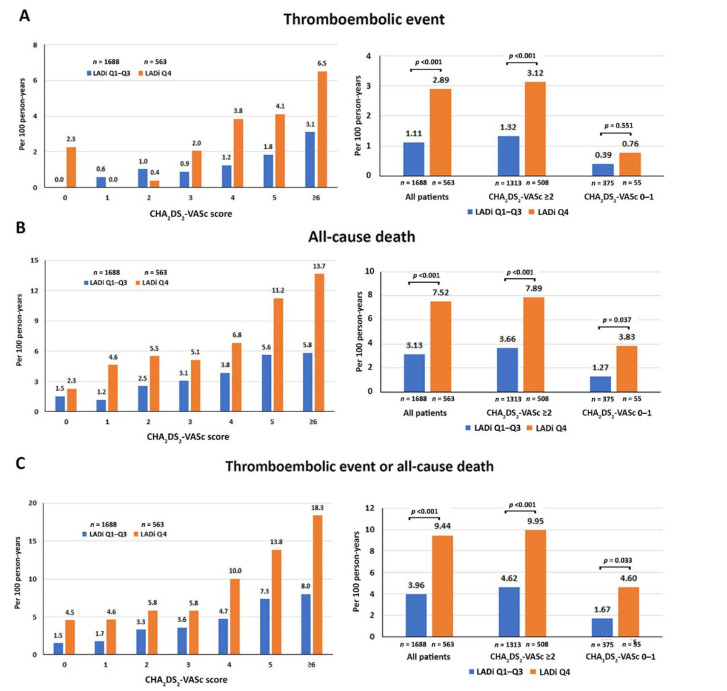
Incidence rate according to combination CHA_2_DS_2_-VASc score and quartiles (Q) of left atrial diameter index (LADi) (left images) and CHA_2_DS_2_-VASc score 0–1 or ≥2 and LADi (right images) for predicting thromboembolic event (**A**), all-cause death (**B**), and thromboembolic event or all-cause death (**C**).

**Figure 3 jcm-11-01838-f003:**
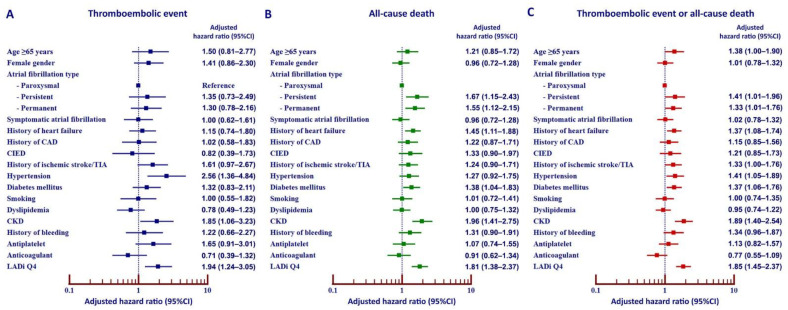
Multivariate analysis to determine the ability of left atrial diameter index (LADi) to independently predict thromboembolic event (**A**), all-cause death (**B**), and thromboembolic event or all-cause death (**C**). Of all the variables included in the analysis, top quartile (Q4) of LADi showed the second highest adjusted hazard ratio for each of the three clinical outcomes.

**Figure 4 jcm-11-01838-f004:**
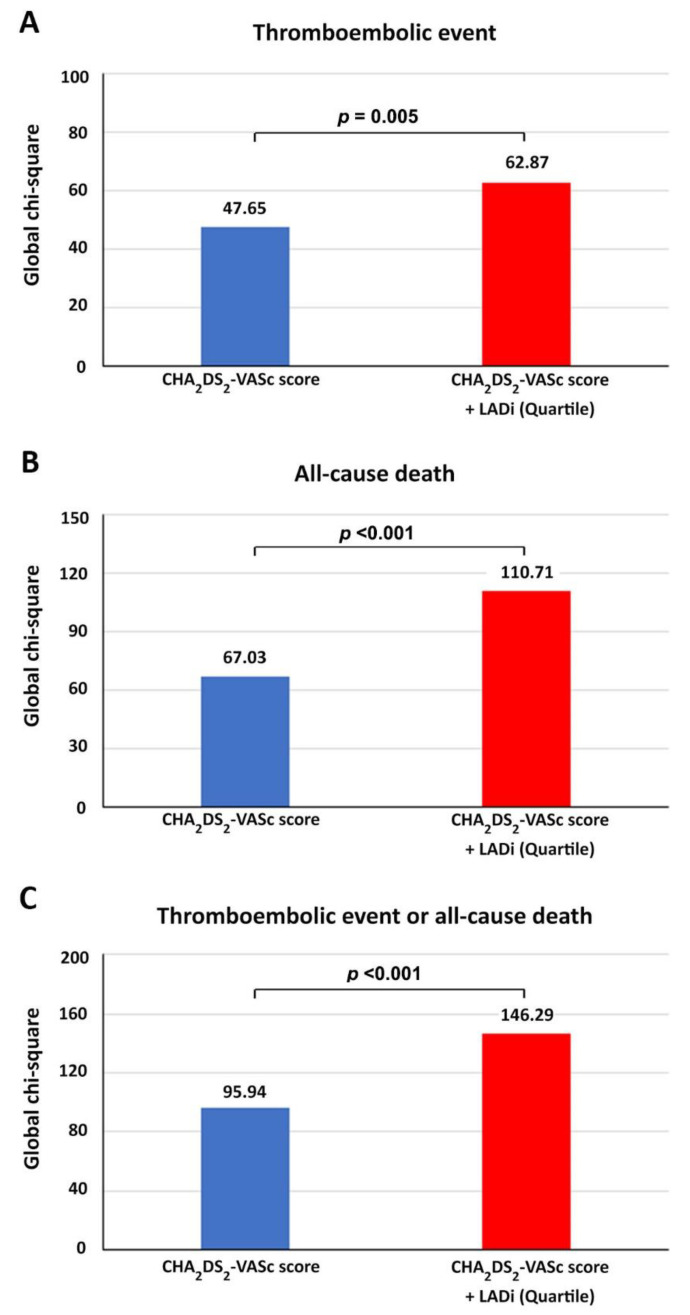
Global chi-square of CHA_2_DS_2_-VASc score and combination CHA_2_DS_2_-VASc score and left atrial diameter index (LADi) for prediction of thromboembolic event (**A**), all-cause death (**B**), and thromboembolic event or all-cause death (**C**).

**Figure 5 jcm-11-01838-f005:**
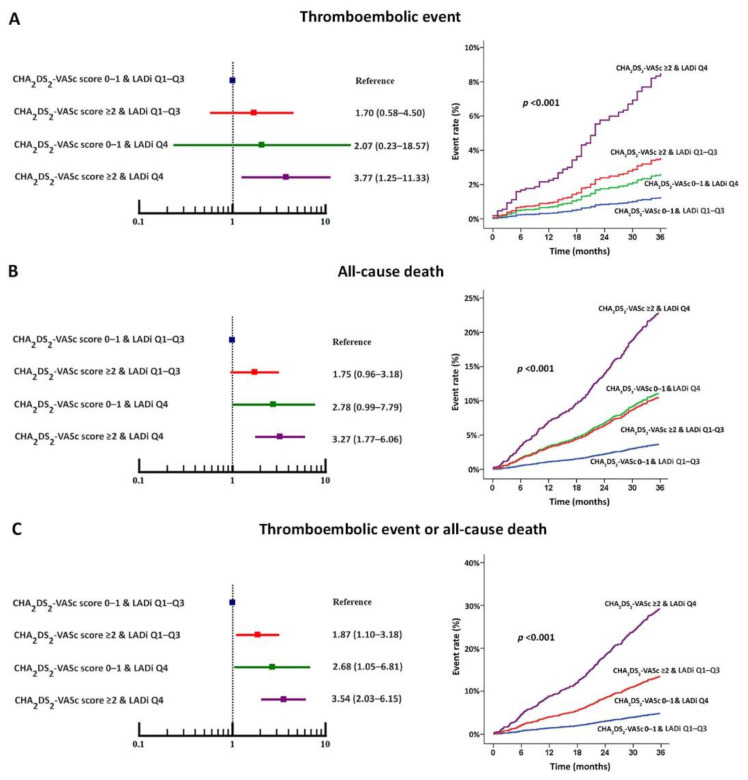
Forest plot (left images) and cumulative incidence of clinical outcomes shown as hazard graph (right images) of thromboembolic event (**A**), all-cause death (**B**), and thromboembolic event or all-cause death (**C**) according to combination CHA_2_DS_2_-VASc score and quartiles (Q) of left atrial diameter index (LADi).

**Table 1 jcm-11-01838-t001:** Baseline characteristics of all patients and compared among the four LADi groups.

Characteristics	All (*n* = 2251)	LADi Q1 (*n* = 563)	LADi Q2 (*n* = 563)	LADi Q3 (*n* = 562)	LADi Q4 (*n* = 563)	*p*-Value
Age (years)	67.38 ± 11.33	63.50 ± 11.87	66.13 ± 10.32	68.19 ± 10.75	71.68 ± 10.72	**<0.001 ^a,b,c,d,e,f^**
Female gender	933 (41.4%)	201 (35.7%)	205 (36.4%)	225 (40.0%)	302 (53.6%)	**<0.001 ^c,e,f^**
Time after diagnosis of AF (years)	3.21 ± 4.12	3.01 ± 4.10	3.00 ± 4.13	3.22 ± 3.94	3.60 ± 4.28	**0.049**
Atrial fibrillation						**<0.001**
Paroxysmal	781 (34.7%)	301 (53.5%)	204 (36.2%)	164 (29.2%)	112 (19.9%)	
Persistent	452 (20.1%)	68 (12.1%)	126 (22.4%)	126 (22.4%)	132 (23.4%)	
Permanent	1018 (45.2%)	194 (34.5%)	233 (41.4%)	272 (48.4%)	319 (56.7%)	
Symptomatic AF	1704 (75.7%)	441 (78.3%)	427 (75.8%)	420 (74.7%)	416 (73.9%)	0.332
History of heart failure	673 (29.9%)	134 (23.8%)	174 (30.9%)	171 (30.4%)	194 (34.5%)	**0.001 ^a,c^**
History of coronary artery disease	369 (16.4%)	89 (15.8%)	91 (16.2%)	95 (16.9%)	94 (16.7%)	0.959
Cardiac implantable electronic device	211 (9.4%)	36 (6.4%)	58 (10.3%)	66 (11.7%)	51 (9.1%)	**0.017 ^b^**
History of ischemic stroke/TIA	381 (16.9%)	97 (17.2%)	80 (14.2%)	95 (16.9%)	109 (19.4%)	0.147
Hypertension	1533 (68.1%)	358 (63.6%)	395 (70.2%)	387 (68.9%)	393 (69.8%)	0.064
Diabetes mellitus	551 (24.5%)	145 (25.8%)	137 (24.3%)	143 (25.4%)	126 (22.4%)	0.546
Smoking	504 (22.4%)	113 (20.1%)	143 (25.4%)	141 (25.1%)	107 (19.0%)	**0.013 ^a,b,e,f^**
Dyslipidemia	1281 (56.9%)	322 (57.2%)	329 (58.4%)	326 (58.0%)	304 (54.0%)	0.427
Renal replacement therapy	27 (1.2%)	0 (0.0%)	10 (1.8%)	4 (0.7%)	13 (2.3%)	**0.002 ^c^**
Dementia	17 (0.8%)	1 (0.2%)	4 (0.7%)	7 (1.2%)	5 (0.9%)	0.216
CKD	1107 (49.2%)	182 (32.3%)	245 (43.5%)	286 (50.9%)	394 (70.0%)	**<0.001 ^a,b,c,e,f^**
History of bleeding	202 (9.0%)	39 (6.9%)	50 (8.9%)	50 (8.9%)	63 (11.2%)	0.099
CHA_2_DS_2_-VASc score						**<0.001**
0	136 (6.0%)	53 (9.4%)	39 (6.9%)	26 (4.6%)	18 (3.2%)	
1	294 (13.1%)	103 (18.3%)	90 (16.0%)	64 (11.4%)	37 (6.6%)	
≥2	1821 (80.9%)	407 (72.3%)	434 (77.1%)	472 (84.0%)	508 (90.2%)	
HAS-BLED score						**<0.001**
0	348 (15.5%)	139 (24.7%)	93 (16.5%)	64 (11.4%)	52 (9.2%)	
1–2	1544 (68.6%)	354 (62.9%)	393 (69.8%)	403 (71.7%)	394 (70.0%)	
≥3	359 (15.9%)	70 (12.4%)	77 (13.7%)	95 (16.9%)	117 (20.8%)	
Antiplatelet	602 (26.7%)	150 (26.6%)	155(27.5%)	169 (30.1%)	128 (22.7%)	**0.047 ^f^**
Anticoagulant	1668 (74.1%)	391 (69.4%)	411 (73.0%)	406 (72.2%)	460 (81.7%)	**<0.001 ^c,e,f^**
Warfarin	1519 (67.5%)	341 (60.6%)	369 (65.5%)	372 (66.2%)	437 (77.6%)	**<0.001 ^c,e,f^**
NOACs	149 (6.6%)	50 (8.9%)	42 (7.5%)	34 (6.0%)	23 (4.1%)	**0.010 ^c^**
Beta blocker	1652 (73.4%)	422 (75.0%)	421 (74.8%)	412 (73.3%)	397 (70.5%)	0.302
CCB—non-dihydropyridine	79 (3.5%)	19 (3.4%)	19 (3.4%)	22 (3.9%)	19 (3.4%)	0.948
Digitalis	387 (17.2%)	68 (12.1%)	98 (17.4%)	94 (16.7%)	127 (22.6%)	**<0.001 ^c^**
MRA	203 (9.0%)	37 (6.6%)	43 (7.6%)	58 (10.3%)	65 (11.5%)	**0.012 ^c^**
Statin	1334 (59.3%)	333 (59.1%)	335 (59.5%)	333 (59.3%)	333 (59.1%)	0.999
ACEI/ARB	1064 (47.3%)	237 (42.1%)	282 (50.1%)	273 (48.6%)	272 (48.3%)	**0.037 ^a^**

Data presented as mean plus/minus standard deviation or number and percentage. A *p*-value < 0.05 indicates statistical significance (bold). ^a^ Statistical significance (*p* < 0.05) LADi Q1 vs. LADi Q2; ^b^ statistical significance (*p* < 0.05) LADi Q1 vs. LADi Q3; ^c^ statistical significance (*p* < 0.05 LADi Q1 vs. LADi Q4; ^d^ statistical significance (*p* < 0.05) LADi Q2 vs. LADi Q3; ^e^ statistical significance (*p* < 0.05) LADi Q2 vs. LADi Q4; ^f^ statistical significance (*p* < 0.05) LADi Q3 vs. LADi Q4. Abbreviations: LADi: left atrial diameter index; Q, quartile; AF, left ventricular atrial fibrillation; TIA, transient ischemic attack; CKD, chronic kidney disease; CHA2DS2-VASc, congestive heart failure, hypertension, age ≥ 75 (doubled), diabetes, stroke (doubled), vascular disease, age 65 to 74 and sex category (female); HAS-BLED, hypertension, abnormal liver/renal function, stroke history, bleeding history or predisposition, labile INR, elderly, drug/alcohol usage; NOACs, non-vitamin K antagonist oral anticoagulants; CCB/NH, calcium channel blocker; MRA, mineralocorticoid receptor antagonists; ACEI/ARB, angiotensin-converting enzyme inhibitors/angiotensin II receptor antagonists.

## Data Availability

The dataset that was used to support the results and conclusion of this study are included within the manuscript. The additional data are available from corresponding author upon reasonable request.

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
