# Peer review of "Left Atrial Diameter in the Prediction of Thromboembolic Event and Death in Atrial Fibrillation"

_jcm, 2022, doi:10.3390/jcm11071838_

Round 1
Reviewer 1 Report
Dear Authors,
The article is interesting.
- Although CHA2DS2-VASc is famous, it is necessary to explain what CHA2DS2-VASc is.
- Line 42: The sentence is too long.
- Line 46-47: it needs more reference, as the ref number 8th studied the case in Thailand.
- Line 47-48: “The thromboembolic complication rate is higher in Asian population than in Western population” needs reference.
Author Response
Response to comments from reviewers
Reviewer 1
Dear Authors,
The article is interesting.
- Although CHA2DS2-VASc is famous, it is necessary to explain what CHA2DS2-VASc is.
Response:
We add explanation for CHA2DS2-VASc on page 5 line 22-23.
- Line 42: The sentence is too long.
Response:
We shortened the statement as suggested.(page 3, line 8-10)
- Line 46-47: it needs more reference, as the ref number 8th studied the case in Thailand.
Response:
We added 2 more references for OAC data from China and India.(page 3, line 14-16)
- Line 47-48: “The thromboembolic complication rate is higher in Asian population than in Western population” needs reference.
Response:
We added 2 references for the referring statement.(page 3, line 16-17)

Reviewer 2 Report
I had a pleasure of reviewing the manuscript entitled: „Left atrial diameter in the prediction of thromboembolic event and death in atrial fibrillation” which concerns the important topic of thromboembolic risk stratification.
I had a pleasure of reviewing the manuscript entitled: „ Assessment of the relationship between anisocytosis and quantitative and qualitative characteristics of coronary atherosclerosis – rationale and study design” which concerns the prognostic value of RDW (red cell distribution width) in patients with coronary artery disease (CAD). Overall, the manuscript concerns an interesting topic. However, there are some major issues that need to be addressed prior to considering the manuscript for publication.
- The manuscript should be corrected by a native-English speaker. There are some major linguistic problems.
- Remove the sentences on warfarin utilisation in your country from the introduction section.
- Elaborate on how the follow-up was conducted
- You should consider adding an additional analysis and dividing patients in accordance with the recommendations of the American Society of Echocardiography, which classify atrial enlargement classified as mild, moderate or severe.
- You cite little number of paper that describe the same issue. You should elaborate this part of discussion eg. doi: 10.1007/s11239-014-1154-6 or doi: 10.1155/2017/6839589
Author Response
Reviewer 2
I had a pleasure of reviewing the manuscript entitled: „Left atrial diameter in the prediction of thromboembolic event and death in atrial fibrillation” which concerns the important topic of thromboembolic risk stratification.
Overall, the manuscript concerns an interesting topic. However, there are some major issues that need to be addressed prior to considering the manuscript for publication.
- The manuscript should be corrected by a native-English speaker. There are some major linguistic problems.
Response:
The manuscript has been edited by the native-English speaker. The English Editor Verification letter is attached in the supplementary material.
- Remove the sentences on warfarin utilisation in your country from the introduction section.
Response:
Since Reviewer 1 suggested us to add more references for warfarin use in other Asian countries in this statement, we have added more references for China and India data. We removed the wording relating to Thailand at the end of the sentence in response to suggestion from Reviewer 2. (page 3, line 14-16)
- Elaborate on how the follow-up was conducted
Response:
We added the following statement to demonstrate how the follow-up was conducted ‘Patients were followed-up at 6, 12, 18, 24, 30, and 36 months. Data relating to cardiovascular events, clinical, laboratory, and medications were collected at each follow-up visit.’(page 5, line 12-13)
- You should consider adding an additional analysis and dividing patients in accordance with the recommendations of the American Society of Echocardiography, which classify atrial enlargement classified as mild, moderate or severe.
Response:
We added a statement on sensitivity analysis of the predictive value of LADi according to the classification of the American Society of Echocardiography in the Results section together with Supplementary Table 1 and Supplementary Figure 2 related to this additional analysis as suggested. (page 10, line 19-22).
- You cite little number of paper that describe the same issue. You should elaborate this part of discussion eg. doi: 10.1007/s11239-014-1154-6 or doi: 10.1155/2017/6839589
Response:
We discussed results of our study in comparison with the results of the 2 suggested papers.(page 13, line 1-6)

Round 2
Reviewer 2 Report
All previously described issues have been sufficiently addressed.